# Myotoxin-3 from the Pacific Rattlesnake *Crotalus oreganus oreganus* Venom Is a New Microtubule-Targeting Agent

**DOI:** 10.3390/molecules27238241

**Published:** 2022-11-25

**Authors:** María Cecilia González García, Caroline Radix, Claude Villard, Gilles Breuzard, Pascal Mansuelle, Pascale Barbier, Philipp O. Tsvetkov, Harold De Pomyers, Didier Gigmes, François Devred, Hervé Kovacic, Kamel Mabrouk, José Luis

**Affiliations:** 1Institut Neurophysiopathol, INP, Faculté des Sciences Médicales et Paramédicales, CNRS, Aix-Marseille Université, 13005 Marseille, France; 2Institut de Microbiologie de la Méditerranée (Marseille Protéomique), IMM (MaP), CNRS, Aix-Marseille Université, 31 Chemin Joseph Aiguier, 13009 Marseille, France; 3Laboratoire LATOXAN SAS, 845 Avenue Pierre Brossolette, 26800 Portes-lès-Valence, France; 4Institut de Chimie Radicalaire, ICR, Faculté des Sciences de Saint Jérôme, CNRS, Aix-Marseille Université, 13397 Marseille, France

**Keywords:** anti-microtubule agent, snake venom, crotamine, tubulin

## Abstract

Microtubule targeting agents (MTA) are anti-cancer molecules that bind tubulin and interfere with the microtubule functions, eventually leading to cell death. In the present study, we used an in vitro microtubule polymerization assay to screen several venom families for the presence of anti-microtubule activity. We isolated myotoxin-3, a peptide of the crotamine family, and three isoforms from the venom of the Northern Pacific rattlesnake *Crotalus oreganus oreganus*, which was able to increase tubulin polymerization. Myotoxin-3 turned out to be a cell-penetrating peptide that slightly diminished the viability of U87 glioblastoma and MCF7 breast carcinoma cells. Myotoxin 3 also induced remodeling of the U87 microtubule network and decreased MCF-7 microtubule dynamic instability. These effects are likely due to direct interaction with tubulin. Indeed, we showed that myotoxin-3 binds to tubulin heterodimer with a Kd of 5.3 µM and stoichiometry of two molecules of peptide per tubulin dimer. Our results demonstrate that exogenous peptides are good candidates for developing new MTA and highlight the richness of venoms as a source of pharmacologically active molecules.

## 1. Introduction

In eukaryotic cells, the microtubule network supports cell shape and polarity and plays a crucial role in several cellular processes, such as cell division, intracellular trafficking, and cell migration. Microtubules are polarized hollow filaments of 25 nm in diameter composed of 13 parallel protofilaments of αβ-tubulin heterodimers. In order to achieve their multiple functions, microtubules need to be robust and yet dynamic polymers [1]. In the cell, the plus end of microtubules is highly dynamic due to the addition (growing or polymerization process) and the removal (shrinking or depolymerization process) of αβ-tubulin heterodimers. Microtubules stochastically alternate between polymerization and depolymerization phases, separated by periods of pause, in a process involving GTP hydrolysis. The switch from microtubule growth or pause to shrinkage is known as “catastrophe,” while the switch from shrinkage to growth or pause is named “rescue” [1].

Microtubule dynamics are finely regulated in the cell by microtubule-associated proteins (MAPs). The binding of MAPs either stabilize (e.g., induce polymerization) or destabilize microtubules (e.g., induce depolymerization), modulating microtubule functions. Exogenous compounds, known as microtubule-targeting agents (MTAs), also bind to microtubules and perturb their dynamics [2]. For example, the taxanes paclitaxel and docetaxel are microtubule-stabilizing agents, while the vinca-alkaloids vinblastine and vincristine are microtubule-destabilizing agents. Because of the essential role of microtubules in cell function, notably in mitosis, MTAs constitute a class of anticancer drugs largely used in clinics [2]. In addition, MTAs have promising therapeutic applications in several other pathologies, including neurodegenerative diseases (for review, see [3,4]). Colchicine, for example, is an MTA used for centuries to treat gout [5].

Most therapeutic molecules come originally from a natural source. Plants have been particularly explored to obtain innovative bioactive compounds, including MTAs. Venoms are also a rich source of bioactive molecules, such as peptides or small molecules. It is estimated that spider venoms contain more than 10 million bioactive peptides [6]. Biomolecules extracted from venoms display very diverse pharmacological activities and can be used as tools for medical research or diagnosis [7]. Several venom-derived molecules have been approved by the Food and Drug Administration and are currently commercialized. For instance, Ziconitide (Prialt^®^) used for the treatment of chronic pain [8] or Exenatide (Byetta^®^) used for the treatment of type 2 diabetes [9] are cone snail and Gila monster toxin-derivatives, respectively. Captopril (Capoten^®^) [10], Eptifibatide (Integrilin^®^) [11], and Tirofiban (Aggrastat^®^) [12] are also drugs based on snake venoms (see ref [13] for a review).

In the present study, we searched for an anti-microtubule peptide. We performed a bioassay-guided screening of about one hundred venoms, from diverse species and families (snakes, scorpions, amphibians, spiders), for their capacity to interfere with microtubule formation. The in vitro tubulin polymerization assay was also used to detect the active compound during the different purification steps. A peptide able to increase tubulin polymerization was isolated from the venom of the Northern Pacific rattlesnake *Crotalus oreganus oreganus*. This peptide, myotoxin-3, is a previously described peptide belonging to the myotoxin crotamine family [14]. Using a combination of biological and biophysical approaches, we showed that myotoxin-3 modifies the microtubule dynamics, resulting in a perturbation of the microtubule network. We also showed that myotoxin-3 binds directly to tubulin with an affinity in the micromolar range. Our results show that snake venom is a valuable source of new MTA candidates.

## 2. Results

### 2.1. Purification of Bioactive Peptides from Venoms

Venoms are known to contain a large quantity of highly potent and selective molecules, mainly peptides and proteins, with diverse pharmacological activities on various targets. The venom from *Crotalus oreganus oreganus* was among the most active on an in vitro microtubule polymerization assay and hence was selected for full identification and characterization of the bioactive compounds. The venom (Figure 1A) was fractionated using preparative RP-HPLC into twenty fractions to facilitate the identification of the active components. Each fraction was tested for its ability to alter tubulin polymerization. All the activity was found in fraction B4 (Appendix A). This fraction was further sub-fractionated (Appendix A), and the activity of each sub-fraction was tested (Appendix A). The active sub-fraction (B4-24) contained two main peaks that were separated by RP-HPLC (Figure 1B). Both peaks contained compounds able to drastically increase the extent of tubulin polymerization and to reduce the lag time before polymerization (Figure 1C) in a dose-dependent manner (not shown).

### 2.2. Mass Spectrometry Characterization

Tubulin polymerization assays showed that both peak 1 and peak 2 contained compounds able to enhance microtubule formation. We thus analyzed the content of both peaks by mass spectrometry (Figure 2A). Matrix-Assisted Laser Desorption Ionization (MALDI) mass spectrometry showed that peak 1 contained a peptide with a molecular weight of 5168.3 Da and a second peptide with a molecular weight of 5144.2 Da. Peak 2 also contained two peptides, one of 4982.4 Da and the second one of 4958.6 Da. It is worth noting that the difference in molecular mass between peptides in each peak was close to 24 Da in both cases.

### 2.3. Amino Acid Sequence Determination

The 5168.3 Da peptide of peak 1 was sequenced by In Source Decay (ISD) MALDI mass spectrometry (Figure 2B). We identified a fragment of 35 amino acids corresponding to the *C*-terminus of the peptide. The annotation showed 100% of local identity with the sequence of myotoxin-3 (UniProt database accession number: P63176), a 45-residue peptide belonging to the crotamine-like peptides found in crotalids. Peptides of the crotamine family are highly cationic at neutral pH and possess six cysteine residues that pair into three disulfide bonds. The remaining moiety of myotoxin-3, i.e., the 10 *N*-terminus amino acids (YKRCHKKGGH), has a calculated monoisotopic mass (Peptide mass calculator) of 1212.69 Da, quite similar to the mass of the last fragment from ISD fragmentation (1212.7 Da) (Figure 2B). The cysteine residues of myotoxin-3 are thought to be engaged in disulfide bonds, although the pairing has not yet been described. The oxidized myotoxin-3 (with the three disulfide bonds) has a molecular mass of 5168.9 ± 1.5 Da, very similar to that (5168.3 Da) of the peptide analyzed by ISD. Altogether, these results clearly demonstrate that the main peptide of peak 1 is myotoxin-3.

The ISD fragmentation spectrum also allowed us to identify a second sequence identical to the previous one except for a shift of -24 Da in its molecular mass (Figure 2B). It is thus possible to suggest that the second peptide present in peak 1 is likely an isoform of myotoxin-3 with a difference of 24 Da in molecular mass.

Peptides from peak 2 were subjected to *N*-terminal Edman sequencing. The sequence obtained was very similar to that of myotoxin-3. The alignment with the sequence obtained by MALDI-ISD (Figure 2C) allowed us to deduce that six out of the seven unidentified residues (X) corresponded to cysteines that cannot be detected because of their engagement in disulfide bonds. The seventh unidentified residue (X38) is very likely a Lysine as this residue is conserved in all published sequences of the crotamine family [15,16]. Moreover, the calculated molecular mass of the sequence with K38 matches the experimental mass perfectly (Figure 3). 

The sequence from peptides in peak 2, obtained by Edman sequencing, was thus identical to that of myotoxin-3 determined by MALDI-ISD, except that (i) the peptide was shorter (43 residues instead of 45) and (ii) the position 5 was occupied by either Histidine or Leucine, with Histidine being the most abundant (Figure 2C). Altogether, the above results suggest that the two peptides present in peak 2 are 43-residue isoforms of myotoxin-3, one with a Histidine at position 5 (calculated mass of oxidized form: 4982.98), the other with Leucine at position 5 (calculated mass of oxidized form: 4959.00). The presence of His instead of Leu results in a difference of 24 Da in the molecular mass of peptides. As this 24 Da difference is also found for the two isoforms in peak 1, we can conclude that the two peptides of peak 1 only differ by the presence of either His or Leu residue at position 5.

In summary, peak 1 contains two 45-residue peptides, i.e., myotoxin-3 (45-residue His^5^-myotoxin-3) and one isoform with Leucine at position 5 (45-residue Leu^5^-myotoxin-3). Peak 2 contains two shorter isoforms (43 residues) of myotoxin-3 with either Histidine (43-residue His^5^-myotoxin-3) or Leucine at position 5 (43-residue Leu^5^-myotoxin-3) (Figure 3). This structure resemblance explains the similarity in their chromatographic behavior. Protein BLAST analysis showed that our four isoforms display high amino acid sequence similarity with other myotoxins (Figure 4). Myotoxin-3 shows high sequence identity with crotamine (86%) and helleramine (90.6%), two myotoxins displaying an anti-proliferative effect.

### 2.4. Chemical Synthesis of Myotoxin-3

We cannot rule out that the activity on tubulin polymerization we measured was due to an undetected contaminant. We thus chemically synthesized myotoxin-3 (45-residue His^5^-myotoxin-3) in order to ascertain that this peptide is indeed responsible for the microtubule-stabilizing activity of the venom. Myotoxin-3 possesses six cysteine residues that are thought to be engaged into three disulfide bonds as indicated by the experimental mass (see above) and as demonstrated for crotamine [17]. After solid-phase synthesis of myotoxin-3 using Fmoc chemistry, the peptide was folded/oxidized using a Tris-ACN solution. The oxidized myotoxin-3 was then purified up to 95% homogeneity by RP-HPLC.

Mass spectrometry analysis of synthetic peptide gave a molecular mass of 5167 Da, demonstrating that all six cysteine residues were oxidized (not shown). When tested in the tubulin polymerization assay, the synthetic myotoxin-3 was able to increase microtubule assembly to the same extent and with a similar efficiency as the native peptide (Figure 5).

The activity of proteins highly relies on their spatial structure. However, for small peptides, this is not necessarily true. It has thus been previously shown that non-structured crotamine (without disulfide bonds) keeps its anti-microbial activity [18,19]. We, therefore, tested whether disulfide bonds were required for the activity of myotoxin-3 on tubulin polymerization. As shown in Figure 5, the non-oxidized form of synthetic myotoxin-3 kept its ability to increase microtubule assembly.

### 2.5. Electron Microscopy of Microtubules

The turbidity enhancement induced by purified and synthetic peptides could also be due to precipitation or aggregation of tubulin. Tubulin was incubated at 37 °C with crude venom or myotoxin-3 as for the tubulin polymerization assay, and samples were analyzed by electron microscopy. Examination showed the presence of microtubules in all samples (Figure 6). The microtubules found in samples with crude venom or with myotoxin-3 (either native or synthetic) were shorter than those found in control. Moreover, in line with the enhanced turbidity, dark regions extremely rich in microtubules could be observed in the presence of venom or peptides.

### 2.6. Intracellular Uptake of Myotoxin-3

Before investigating whether myotoxin-3 was able to affect the microtubule cytoskeleton, we verified that peptide could penetrate the cell. Native 45-residue myotoxin-3 was thus covalently conjugated to the cyanine dye Cy3. Prior to incubation with labeled peptide, MCF-7 cells were transfected with a farnesyl-GFP plasmid to visualize the cell membrane. In order to limit the fluorescence signal to cell-internalized peptide, cells incubated with Cy3-myotoxin-3 were treated at low pH to remove cell surface-bound peptide. Cy3-myotoxin-3 efficiently penetrated the cell, as fluorescence could be observed within the cell after only a 5 min incubation (not shown). After 1h, most cells were stained to a more or less extent (Figure 7), with some cells displaying a uniform and intense Cy3-myotoxin-3 signal (arrowhead). Myotoxin-3 was found mostly in the cytosol and was concentrated in dots distributed around the nucleus. Note that a z-stack of images confirmed that Cy3-labeled myotoxin-3 was restricted within the cell (Video S1). The same observations were made when using the synthetic peptide (data not shown).

### 2.7. Myotoxin-3 Decreases Microtubule Dynamics in Living Cells

Microtubule dynamics are critical for diverse cellular functions. Since myotoxin-3 affected tubulin polymerization in vitro, we investigated whether the peptide was able to interfere with microtubule dynamics in living MCF-7 cells transiently expressing GFP-tagged tubulin. We analyzed the dynamic instability of individual microtubules within the lamellar region (Figure 8A). Cells were incubated for 3h with native myotoxin-3 at various concentrations, and microtubules were monitored by video-microscopy to determine the parameters of microtubule dynamic instability. The microtubules plus ends of the lamellar region alternated between phases of slow-growing, rapid shortening, and prolonged pause state (a state of attenuated dynamics) (see Appendix A). 

Figure 8B shows the life history changes in the length of microtubules in the absence (control) or in the presence of native 45-residue myotoxin-3 at various concentrations. Interestingly, upon treatment with myotoxin-3, we observed a dose-dependent smoothing of microtubule end trajectories with less extensive growth and shortening events compared with control cells, suggesting that the peptide reduced microtubule dynamics. This was confirmed by the calculation of the overall microtubule dynamicity, which decreased in a dose-dependent manner in the presence of peptides (Figure 8C and Table 1). 

The microtubule dynamics parameters obtained for control MCF-7 cells showed characteristic highly dynamic microtubules, and significant differences in several parameters were observed in a concentration-dependent manner upon treatment with peptide (Table 1). The microtubule growth and shortening rates decreased by 21% upon treatment with 25 µM myotoxin-3. The effect on growth and shortening lengths was even more pronounced, with a reduction in 40% and 25%, respectively. We also observed a decrease in the percentage of time spent in growth (−29%) and an increase in the percentage of time spent in pause (+15%) and in the time-based transition frequencies of catastrophes (+43%) and rescues (+70%). This overall resulted in the reduction in the global microtubule dynamicity (−40%) from 14.4 ± 1.73 µm/min to 8.65 ± 0.68 µm/min (Table 1).

### 2.8. Myotoxin-3 Affects Microtubule Network of U87 Cells and Induces Cytotoxicity

Considering that myotoxin-3 affected microtubule dynamics, we investigated whether the peptide was able to also disturb the organization of the microtubule and actin cytoskeleton. The effect was evaluated by immunofluorescence microscopy on MCF-7 and U87 cell lines after 24 h treatment with increasing concentrations of native 45-residue myotoxin-3 (Figure 9).

Untreated cells displayed microtubules filling the whole cytoplasm up to the edges of the cell. Treatment of MCF-7 cells with myotoxin-3 did not alter either microtubules or actin cytoskeleton, even at a concentration of 50 µM (Figure 9A). On the other hand, the microtubule network was greatly affected in U87 cells (Figure 9B). Indeed, treatment with 1 µM myotoxin-3 induced remodeling of the microtubule network into buckled bundles in some cells (Figure 9B, arrow and inset), while the actin fibers seemed unaffected. The number of cells with a wavy pattern increased with peptide concentration. Highest concentrations of myotoxin-3 induced numerous cells to lose their elongated morphology and to round up. 

To test the cytotoxicity of myotoxin-3, we incubated U87 and MCF-7 cells in the presence of increasing concentrations of synthetic peptides. After 24h, cell viability was quantified by MTT assay. As illustrated in Figure 9C, the peptide induced a moderate dose-dependent death of both U87 and MCF-7 cells. Interestingly, MCF-7 cells were less sensitive to myotoxin-3 treatment than U87 cells, as observed above in the case of the microtubule network.

### 2.9. Myotoxin-3 Binds to Microtubules and Tubulin

As described above, myotoxin-3 increased microtubule assembly. We thus next checked whether our peptide binds to microtubules in vitro. The microtubules obtained in the presence of 1 µM native 45-residue myotoxin-3 were sedimented by ultracentrifugation, and pellets were washed with buffer at 37 °C to remove unbound peptide. After depolymerization of microtubules by decreasing temperature, peptides smaller than 30 kDa were isolated from tubulin and analyzed by mass spectrometry. MALDI MS revealed the presence of a peptide of 5168 Da corresponding to myotoxin-3 (Figure 10A), demonstrating that the peptide co-sedimented with microtubules.

Co-sedimentation experiments described above suggest that myotoxin-3 interacts with microtubules and hence with tubulin. We thus investigated the association of myotoxin-3 with αβ-tubulin dimers by analytical ultracentrifugation at 20 °C in a buffer that does not allow microtubule assembly at the tubulin concentration used. As expected for 10 μM tubulin alone, we observed a single Gaussian distribution of the continuous sedimentation coefficient, C(S), indicating tubulin sediments as a single species (Figure 10B). The C(S) centered at 5.3 ± 0.9 S (Figure 10B, left panel). In the presence of 10 μM native 45-residue myotoxin-3 and 10 µM tubulin, we also observed a single Gaussian distribution of the C(S), with a value of the sedimentation coefficient increased to 5.9 ± 0.8 S. (Figure 10B, left panel). The same result was obtained when using 43-residue myotoxin-3, with C(S) values of 5.8 ± 0.6 S and 6.3 ± 0.9 S for tubulin alone and tubulin plus peptide, respectively (Figure 10B, right panel). The increase in the sedimentation coefficient of tubulin in the presence of peptide suggests a direct interaction between myotoxin-3 and ab-tubulin dimers.

To obtain access to the parameters of binding of myotoxin-3 to ab-tubulin dimers, we used isothermal titration calorimetry (ITC). This biophysical technique needs amounts of myotoxin-3 that can hardly be obtained by purification. Even the yield of chemical synthesis is hindered by the presence and the formation of the three disulfide bonds in the right place. Having shown that non-oxidized myotoxin-3 (without disulfide bonds) was as efficient as the native peptide in favoring microtubule formation (see Figure 5), we performed ITC using this form of mytoxin-3. Titration of 55 µM tubulin by 2.2 mM of myotoxin-3 resulted in an exothermic binding curve, showing that this is an enthalpy-favorable interaction. Fitting experimental data with the “one-set-of-sites” model (Figure 10C) revealed that tubulin is able to bind two myotoxin-3 molecules per tubulin heterodimer with low micromolar dissociation constant (Kd = 5.3 µM) in the same range as other MTAs [20]. We can thus hypothesize that each highly homologous a tubulin and b tubulin subunit binds one myotoxin-3 molecule.

## 3. Discussion

Venoms are designed to immobilize prey by disrupting the biological processes and the vital functions of organs. They are complex mixtures of pharmacologically active components, including proteins, peptides, and enzymes with specific biological activities, as well as other organic compounds of low molecular weight, ions, and inorganic substances [21]. Venoms are thus precious sources of bioactive components with various valuable activities [7,21]. We previously isolated from the *Argiope lobata* spider venom a compound able to regulate melanin production through the inhibition of tyrosinase [22]. In the present paper, we show for the first time that snake venom contains microtubule-targeting activity. Indeed, from *C. o. oreganus* venom, we isolated myotoxins capable of binding tubulin and increasing microtubule assembly in vitro. The peptides also affected the microtubule network and dynamics in the cell. One of these peptides, myotoxin-3, was previously isolated and described in 1990 from *Crotalus viridis viridis* venom [14]. The three other peptides are isoforms of myotoxin-3 and differ from myotoxin-3 in the number of residues and in the presence of either Histidine or Leucine at position 5 (45-residue Leu^5^-myotoxin-3, 43-residue His^5^-myotoxin-3 and 43-residue Leu^5^-myotoxin-3). The 43-residue His^5^-myotoxin-3 isoform, called peptide C, was isolated in 1978 from *Crotalus viridis helleri* venom [23].

We showed here that myotoxin-3 facilitates in vitro microtubule assembly. Indeed, both native isoforms and synthetic myotoxin-3 were able to reduce the lag time before tubulin polymerization and to increase the extent of microtubule assembly. As all peptides of the crotamine family, myotoxin-3 and its three isoforms possess six cysteine residues that form three disulfide bonds, as confirmed by our experimental molecular masses obtained by mass spectrometry. However, the disulfide bonds are not required for the induction of polymerization as the non-oxidized form (without disulfide bonds) of myotoxin-3 was able to enhance microtubule polymerization as well. This suggests that the tubulin-targeting activity depends on the primary structure of myotoxin-3 rather than on its tridimensional structure. In line with this finding, it has previously been reported that DTT-reduced crotamine and non-structured recombinant crotamine expressed in bacteria kept their antimicrobial activity [18,19]. 

To be useful, MTAs should have access to the microtubular network and hence penetrate within the cell. As reported for crotamine [24], myotoxin-3 possesses cell-penetrating properties, as we observed a translocation of Cy3-labeled peptide into the cell. Moreover, myotoxin-3 altered the dynamics of microtubules in MCF-7 cells (Table 1). Upon treatment with peptide, we thus noticed variations in most microtubule dynamics parameters that resulted in a diminution of 40% of the overall microtubule dynamicity with 25 µM myotoxin-3. Disturbing the dynamic instability of microtubules is a hallmark of microtubule-targeting agents that act by either stabilizing or destabilizing microtubules (for review, see [2,4]). The reduced microtubule dynamics in the presence of myotoxin-3 are consistent with the enhanced microtubule formation observed in in vitro assays. 

Although myotoxin-3 penetrated into MCF-7 cells and altered the microtubule dynamics in cultured cells, this did not result in a visible alteration of the microtubule network in this cell line. It is possible that myotoxin-3 reduced microtubule dynamics without noticeably perturbing the microtubule network, as already observed with some other MTAs [25]. On the contrary, myotoxin-3 markedly affected the organization of the microtubular network in U87 cells. Treatment of cells with myotoxin-3 for 24h led to a dose-dependent remodeling of microtubules into wavy bundle-like structures. With higher concentrations, we observed a reduction in cell area and, in some cases, cell body shrinkage. It has been shown that the light chains of MAP1a, the most abundant MAP in the adult brain, promote tubulin polymerization [17], stabilize microtubules, and reorganize them into wavy bundles [26,27]. Interestingly, light chains increase amounts of detyrosinated microtubules that colocalized with wavy microtubules [27]. As detyrosinated tubulin is abundant in the brain and is carried by stable microtubules [28], it could thus be interesting to verify whether the stabilization of microtubules induced by myotoxin-3 in glioblastoma U87 cells is associated with detyrosination of tubulin and whether detyrosinated microtubules are located into wavy bundles.

We also observed a different response of MCF-7 and U87 cells when measuring cell viability upon myotoxin-3 treatment. Although cytotoxicity was modest, the destabilization of the microtubule network was associated with a higher sensitivity to the treatment of U87 cells compared to MCF-7 cells. It is interesting that MCF-7 cells do not behave like U87 cells concerning their sensitivity to myotoxin-3. The tubulin isotype composition impacts the structural properties of microtubules and tunes their dynamics. Tubulin is also subjected to a large diversity of post-translational modifications that appear to program microtubules for specific functions. The concept that tubulin isotypes and post-translational modifications allow microtubules to display different structural and functional properties between cell types or even within the cell is known as the ‘tubulin code’ (see [29] for an extensive review). In this regard, it is worth noting that the tubulin used for the in vitro screening assays was purified from the brain. Myotoxin-3 may thus preferentially target a tubulin isotype predominantly expressed in neurons, which could explain the higher sensitivity of glioblastoma U87 cells compared to epithelial MCF-7 cells. Such a relative resistance of MCF-7 cells to anti-microtubule agents has already been reported [30]. The authors observed that, unlike HeLa cells that detached and died, vincristine-treated MCF-7 cells reattached to the culture flask after a failed mitosis and remained viable. The resistance of MCF-7 could thus be due to the expression of a particular tubulin isotype combination or to post-translational modifications that make cells less sensitive to myotoxin-3. A possible candidate could be the β3-tubulin isotype. Indeed, although glioblastoma do not derive from neurons, Abbassi et al. [31] reported the presence of the neuronal β3-tubulin isotype in U87 cells, while MCF7 cells predominantly express β4-tubulin isotype (58.4%) and in a much lesser extent β3-tubulin (2.5%) [32]. 

Little is known about the effect of venom peptides on the microtubule cytoskeleton. Toad venoms have been reported to destabilize microtubule network organization [33]. However, this effect is likely not direct, as the major active compounds, such as the bufadienolides proscillaridin A and bufalin, act by affecting microtubule dynamics through the phosphorylation of EB1 by GSK3 [34]. The phospholipase A2, MVL-PLA2, and the Kunitz-type serine protease inhibitor, PIVL, from viper venom also increase microtubule dynamic instability in cultured cells [35,36]. The effect is, there as well indirect as these toxins interact with integrins located at the outside of the cell [35,36]. 

The effect of myotoxin-3 on the microtubule network is likely due to direct interaction with microtubules. Indeed, we showed that the peptide binds to tubulin heterodimer with a Kd of 5.3 µM, a value similar to other MTAs [20], and a stoichiometry of two molecules of myotoxin-3 per ab-tubulin dimer. On one hand, each α and β subunit of tubulin possesses an unstructured C-terminal tail with a high proportion of glutamates that is exposed at the outside surface of the microtubule. On the other hand, myotoxin-3 is a cationic peptide displaying nine lysine and two arginine residues. It is known that the electrostatic interaction with the C-terminal is employed by endogenous microtubule-associated proteins, such as Tau and MAP7, to promote microtubule assembly and stability [37,38]. It would thus be interesting to check whether the two binding sites of myotoxin-3 on the ab-tubulin dimer could interact with the two anionic C-terminal tails and whether this interaction is responsible for the increase of microtubule polymerization. 

Most MTAs are natural compounds derived from bacteria, plants, fungi, and sponges [39]. In this work, we demonstrate that venoms are also valuable natural sources of bioactive molecules targeting microtubules. Although further studies should be carried out to unveil the mechanism of action of myotoxin-3, this peptide represents a new MTA candidate isolated from an animal venom.

## 4. Materials and Methods

### 4.1. Reagents

Kitoxan^®^ and venoms, obtained from animals in captivity, were kindly donated by Latoxan (Portes-lès-Valence, France). Venoms were provided as lyophilized venoms and kept at −20 °C prior to analysis and at −80 °C after solvation. The synthetic reduced myotoxin 3 was purchased from Genecust (France), and the native crotamine was purchased from Latoxan (Portes-lès-Valence, France). All HPLC-grade solvents were from Sigma-Aldrich (Saint-Quentin-Fallavier, France). The chemicals and reagents for tubulin purification and assays were a molecular grade or the highest purity available from Sigma-Aldrich and human plasma fibronectin from Millipore (Temecula, USA).

### 4.2. Purification of Bioactive Peptides from C. o. oreganus Venom

Lyophilized crude snake venom from *C. o. oreganus* (0.05 g) was reconstituted on water, centrifuged at 9600 rpm for 5 min, filtered (0.45 µm), and then applied to a reverse-phase C18 preparative column (Phenomenex Jupiter 5u C18 250x21.2) equilibrated with 0.1% TFA in water (solvent A). The column was eluted at a flow rate of 6 mL/min with 90% acetonitrile, 9.9% water, and 0.1% TFA (solvent B), using a three-step gradient (0–25% of solvent B for 60 min; 25–50% solvent B for 100 min; and 50–100% of solvent B in 100 min). Fractions were collected, lyophilized, and tested for tubulin polymerization. 

The fraction showing a perturbation of tubulin polymerization was then subfractionated, and the tubulin polymerization assay was performed for each of these subfractions. For further analysis, the active subfractions were applied to a reverse-phase C18 analytic column (RP-Nucleodur 100-5 4.6 × 250 mm) equilibrated with 0.1% TFA in water and then eluted at a flow rate of 1 mL/min with a linear gradient 20–40% B solvent over 40 min. Active fractions were lyophilized and stored as a water solution at −20 °C until use.

### 4.3. In Vitro Tubulin Polymerization Assay

Lamb brains were obtained from freshly slaughtered animals, kept on ice, and used within 1 h after their death. Lamb brain tubulin was purified by ammonium sulfate fractionation and ion-exchange chromatography [40]. Tubulin was stored on sucrose buffer in liquid nitrogen until use. For polymerizing assays, tubulin was thawed and equilibrated in assembly buffer (0.1 mM GTP, 10 mM MgCl_2_, 100 mM EGTA, 20 mM phosphate buffer, and 3.4 M glycerol, pH 6.5) by chromatography on a Sephadex G25 column. Tubulin concentration in 6 M guanidine hydrochloride was determined spectrophotometrically at 275 nm (ε^M^ = 109,000 M^−1^ cm^−1^) as previously [41]. Tubulin at a final concentration of 10 µM was mixed with venom or peptides at the indicated concentration and immediately placed into the spectrometer. The mixture was heated from 15 °C to 37 °C, and the absorbance was followed at 350 nm.

### 4.4. Electron Microscopy of Microtubules

Tubulin was allowed to polymerize for 30 min at 37 °C, alone or with peptides. Samples were then adsorbed onto 200 mesh, Formvar carbon-coated copper grids, stained with 2% (*w/v*) uranyl acetate, and let to dry. Grids were observed with a Jeol 100C transmission electron microscope to confirm the presence of microtubules.

### 4.5. Molecular Mass Determination by MALDI-MS

Matrix-Assisted Laser Desorption Ionization-Mass Spectrometry (MALDI-MS) spectra were obtained either on an UltraflXtreme apparatus (BrukerDaltonics, Bremen, Germany) operating in positive linear mode with delayed extraction or on a MicroflexII apparatus (BrukerDaltonics, Bremen DE) in positive reflectron mode. The samples were co-crystallized with a 10 mg/mL solution of hydroxy α-cyano-4-cinnamic acid (HCCA) on the MALDI-MS target by the dry droplet method. MALDI-MS spectra were acquired with an accelerating potential of 20 KV and a laser power set to the minimum level necessary to obtain an efficient signal. Spectrum mass calibration was based on external calibration using an appropriate peptide standard mixture (Peptide Calibration Standard, BrükerDaltonics).

### 4.6. Amino Acid Sequence Determination

MALDI in-source decay was performed using an Ultraflextreme TOF/TOF mass spectrometer controlled by the FlexControl 3.3 software (BrukerDaltonics). Spectra were acquired in positive reflectron ion mode with 2000 laser shots accumulated, and the laser power was set with an increase in 20% of fluensce to fragment protein in the source of the mass spectrometer using a 20 mg/mL solution of 1,8-diaminonaphtalen (DAN) matrix enabling the generation of hydrogen radicals that break the peptide backbone producing c-ions and z-ions from 1000 to 5000 Da. The generated fragment ions allowed the sequence annotation of the peptides. The mass spectrometer parameters were set according to the manufacturer’s settings for optimal acquisition performance. Sequences were analyzed on Flex Analysis software, version 3.0 (BrukerDaltonics). Mass spectrometer calibration was performed using the c-ions of 1 pmole of BSA spotted with the DAN matrix.

*N*-terminal sequencing of the peptide present on the active fraction was performed by Edman degradation. Approximately 600 pmol of the sample was loaded on a glass fiber membrane pretreated with 10 µL of biobrene and then analyzed using an automatic sequencer model PPSQ 31B (Shimadzu, Kyoto, Japan). The sequence was compared with the Uniprot database and with the sequence obtained by MS.

### 4.7. Chemical Synthesis of Myotoxin-3

The 45-residue sequence YKRCHKKGGHCFPKTVICLPPSSDFGKMDCRWKWKCCKKGSVNNA-COOH was obtained by the solid-phase method using a peptide synthesizer (Model 433A, Applied Biosystems). The synthetic myotoxin-3 was incubated in a 0.1 M Tris (pH 8.3)-ACN (70-30, *v/v*) solution to allow the oxidation of the six cysteines and the formation of three disulfide bonds. The oxidized peptide was then purified by RP-HPLC, and the oxidation was confirmed by comparing the mass by MALDI-TOF of the non-oxidized synthetic peptide obtained by MALDI-TOF versus the oxidized synthetic peptide.

### 4.8. Intracellular Uptake of Myotoxin-3

Native and synthetic non-oxidized myotoxin-3 were labeled with Cy3 fluorescent dye (Cy3 monoNHS ester, Amersham, UK) accordingly with fabricant instructions. Briefly, the peptide was dissolved in Na_2_CO_3_/NaHCO_3_ buffer at pH 9.3, while the Cy3 dye was dissolved in DMSO. The peptide and the Cy3 fluorescent dye were incubated for 45 min. The labeled peptide was separated from free Cy3 dye by centrifugation in a filter device with a cutoff of 3 kDa (Amicon^®^ Ultra, Merck), Fontenay-sous-Bois, France.

Cells were routinely maintained at 37 °C and 5% CO_2_ in EMEM medium (Gibco, Cergy-Pontoise, France) containing 10% fetal bovine serum. Adherent MCF-7 cells seeded in 10 µg/mL fibronectin-coated slides were transfected with 0.3 µg of farnesyl-GFP DNA plasmid kindly provided by Dr. Papandréou using Lipofectamine^TM^ 3000 (Thermo Fisher Scientific, Waltham, USA) and Opti-MEM^®^ (Gibco) following the fabricant recommendations. After 24 h of transfection, MCF-7 cells were incubated with labeled Cy3-myotoxin-3 or PBS (vehicle) in EMEM supplemented with 5% fetal calf serum and penicillin/streptomycin under 5% CO_2_, at 37 °C and 90% humidity, for 1 h. Cells were incubated at low pH (0.2 M acetic acid, 0.5 M NaCl) for 8 min at 4°C to remove cell surface bound peptide [42], rinsed with ice-cold PBS, fixated in 4% paraformaldehyde-0.2% glutaraldehyde and mounted onto slides for observation. Images were acquired using a Leica SP5 confocal laser scanning microscope (CLSM) with a Leica inverted microscope equipped with a Plan-Apochormat 63× 1.4 oil immersion objective (NA = 1.4). Each image was recorded with the spectral mode of the CLSM selecting specific domains of the emission spectrum: the secondary AlexaFluor488-conjugated antibody was excited at 488 nm with an argon laser, and its fluorescence emission was collected between 496 nm and 535 nm, and the TRITC-coupled phalloidin was excited at 543 nm with a helium-neon laser and its fluorescence emission was collected at between 580 nm and 610 nm. The two fluorophores were excited sequentially. The public-domain ImageJ software was used for image analysis.

### 4.9. Time-Lapse Microscopy and Analysis of Microtubule Dynamics

MCF-7 cells were seeded at a density of 1.5 × 10^4^ per well in 8-well Nunc Lab-Tek II Chamber Slides™ (Thermo Fisher Scientific) coated with 10 µg/mL fibronectin. Adherent cells were then transfected with 0.3 µg of tubulin-GFP DNA plasmid and incubated for 24 h. MCF-7 cells were treated with myotoxin-3 (0, 1, 5, or 25 µM) in observation media (phenol-free media, 1% FCS, 10 mM HEPES pH7.4, penicillin/streptomycin) for 3 h at 37 °C under 5% CO_2_ and 90% humidity. Time-lapse acquisitions for microtubule tracks were performed at 2-s intervals for 1 min by CLSM. For calculating dynamics, 50 to 80 individual microtubule trajectories were followed using manual tracking plugging from ImageJ software. Statistical differences were calculated using GraphPadPrism.

### 4.10. Microtubules Labeling

MCF7 and U87 cells were seeded at a density of 15 × 10^3^ and 30 × 10^3^ cells per well, respectively, in coverslips coated with 10 µg/mL fibronectin and cultured for 24 h. Cells were treated for 24 h with myotoxin 3 at different concentrations in EMEM media supplemented with penicillin/streptomycin. Cells were then rinsed with PBS, fixated with 4% paraformaldehyde-0.2% glutaraldehyde in PBS, and permeabilized with Triton-X100. Permeabilized cells were immunostained with 1 µg/mL anti-α-tubulin antibody (clone D1MA, Sigma), followed by the secondary Alexa488-conjugated antibody at 5 µg/mL (Thermo Fisher Scientific). Actin was subsequently stained with 1µg/mL phalloidin-tetramethyl rhodamine B isothiocyanate (Sigma). Coverslips were mounted for observation with Prolong Gold antifade (Thermo Fisher Scientific), and images were acquired using a Leica SP5 confocal laser scanning microscope (CLSM) equipped with a Plan-Apochromat 63x 1.4 oil immersion objective.

### 4.11. Cytotoxicity Assay

Cell viability was assessed by MTT (3-(4,5-dimethylthiazol-2-yl)-2,5-diphenyltetrazolium bromide) assay. U87 cells were seeded in a microtiter plate (5000 cells/well) and incubated overnight at 37 °C in a humidified atmosphere and 5% CO_2_. The medium was renewed and supplemented with peptides to be tested in the presence of penicillin/streptomycin. After 24 h, cells were incubated with a culture medium containing 0.5 mg/mL MTT at 37 °C for 3 h. After removing the MTT solution, the precipitated formazan crystals were solubilized with 100 μL DMSO and absorbance was measured at 550 nm.

### 4.12. Co-Sedimentation Microtubules/Myotoxin-3

Tubulin was allowed to polymerize for 30 min. at 37 °C in the presence of native myotoxin-3. Microtubules were sedimented at 30,000× *g* for 10 min at 37 °C and washed twice at 37 °C with 250 µL polymerization buffer (20 mM phosphate buffer, pH 6.5, 0.1 mM GTP, 10 mM MgCl2, 100 mM EGTA, and 3.4 M glycerol). The pellet was then resuspended in 20 mM phosphate buffer, pH 6.5 at 4 °C to depolymerize microtubules. Peptides were separated from tubulin dimers by ultrafiltration on Vivaspin 30 kDa MWCO columns (1200× *g* for 8 min. at 4 °C), concentrated on a C18 column, and analyzed by mass spectrometry.

### 4.13. Sedimentation Velocity

Experiments were carried out at 40,000 rpm and 20 °C in a Beckman Optima-XL-A analytical ultracentrifuge, using 3 mm double sector centerpieces in an AN50Ti rotor. Scans were acquired in continuous mode at 275 nm. At 20 °C, the partial specific volume of tubulin was 0.736 mL.g^−1^. The buffer was composed of 20 mM NaPi, 10 μM GTP, pH 6.5, and its density and viscosity were 1.009 g/mL^−1^ and 0.0147 poise, respectively. All these parameters were calculated with SEDNTERP designed by J Phylo [43]. The data recorded from moving boundaries were analyzed in terms of continuous size distribution functions of sedimentation coefficient, C(S) using the program SEDFIT [44], and the apparent sedimentation coefficient at 20 °C in water (S_20,W_) determined by peak integration.

### 4.14. Isothermal Titration Calorimetry

The binding of myotoxin 3 to tubulin was studied on a PEAQ-ITC isothermal titration calorimeter at 20 °C, in a non-assembly buffer (20 mM sodium phosphate pH 6.5, 1 mM TCEP, and 0.1 mM GTP). The tubulin concentration in the calorimetric cell was 55 µM, whereas the concentration of myotoxin 3 in the titrating syringe was 2.2 mM. The volume of the injections was set to 2 µL. The dilution heat was measured by titration of the myotoxin 3-containing buffer into the same buffer without the protein. The obtained curve was analyzed by instrument software using the “one-set-of-sites” model. Therefore, the stoichiometry (N), constant (Ka), enthalpy (ΔH), and entropy of binding (ΔS) were calculated from the standard thermodynamic equations, as previously described [45].

## Figures and Tables

**Figure 1 molecules-27-08241-f001:**
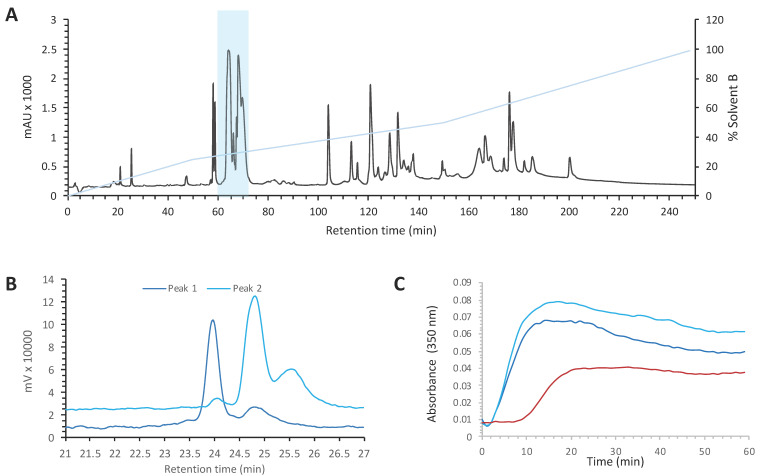
Purification of myotoxin-3 from *Crotalus oreganus oreganus* venom. (**A**) Preparative HPLC chromatogram of *C. o. oreganus* venom. The position of the fraction B4 is highlighted (**B**) Superposed profiles (214 nm) of peaks present in the sub-fraction B4-24, peak 1 (dark blue) and peak 2 (light blue), loaded separately on analytical RP-HPLC column. (**C**) Tubulin polymerization assay in the absence (red line) or in the presence of peak 1 (dark blue) or peak 2 (light blue). The increase in turbidity at 350 nm indicates polymerization of the αβ-tubulin dimer.

**Figure 2 molecules-27-08241-f002:**
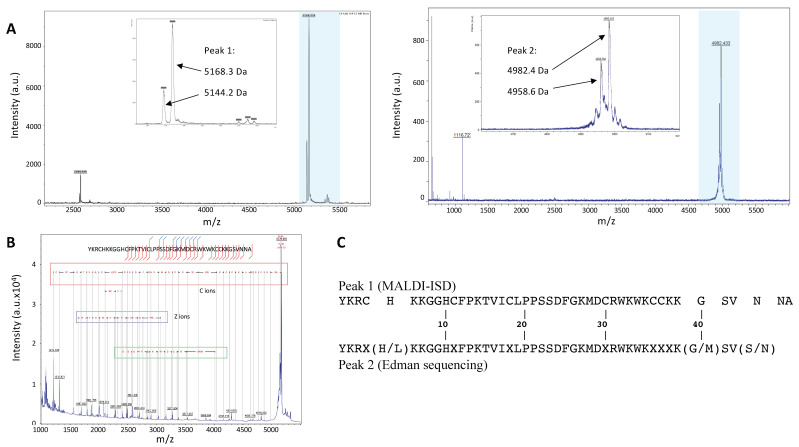
Mass spectrometry characterization and amino acid sequence determination. (**A**) Peptide masses values in peak 1 (left) and peak 2 (right) were determined by MALDI-TOF. The highlighted regions are widened in the insets. (**B**) The amino acid sequence of peptides from peak 1 was determined by ISD. (**C**) Alignment of the main peptide sequence of peak 1 (by MALDI-ISD) with that obtained by Edman degradation for peptides in peak 2.

**Figure 3 molecules-27-08241-f003:**
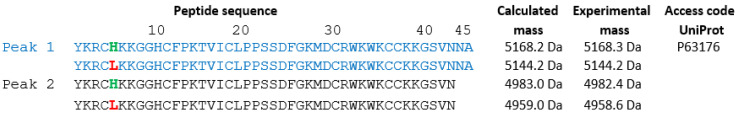
The amino acid sequence of myotoxin-3 and its isoforms. The amino acid sequence of peptides from peak 1 (blue) and peak 2 (black) with calculated (ProtParam tool) and experimental masses. Residues in position 5 residues are highlighted in bold. Myotoxin-3 corresponds to the first peptide (UniProt database accession number P63176).

**Figure 4 molecules-27-08241-f004:**
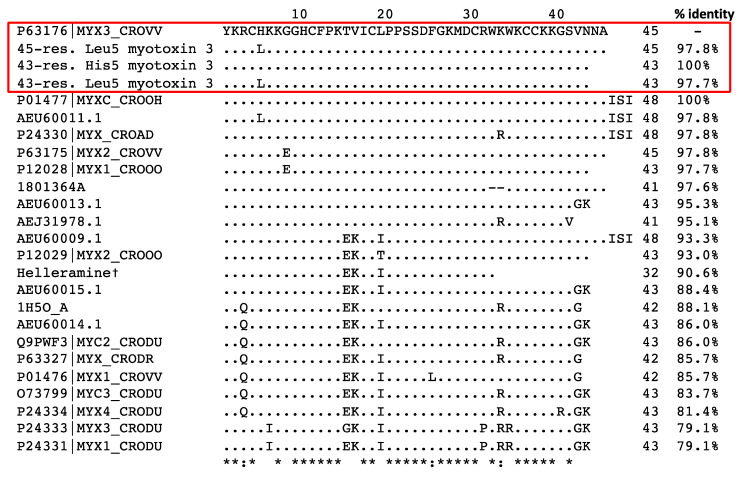
Alignment of the amino acid sequence of myotoxin-3 with homologous peptides. The amino acid sequences from this paper are in a red box. †: Salazar et al. 2020 (ref. [15]). The other sequences were obtained from the UniProtKB/SwissProt database. Dots indicate identities to myotoxin-3 (P63176|MYX3_CROVV), asterisks (*) indicate identical amino acid residues and the symbols “:” indicate conserved substitutions.

**Figure 5 molecules-27-08241-f005:**
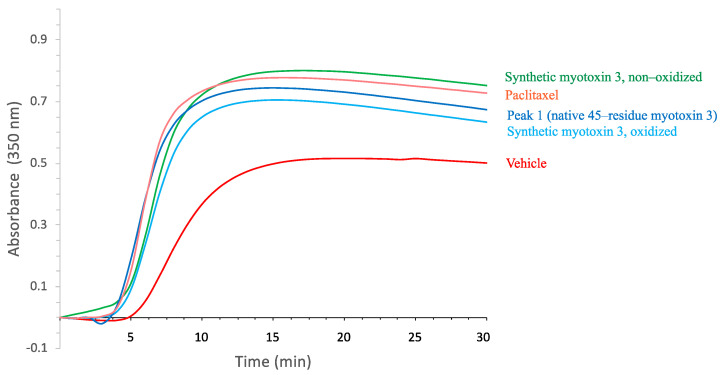
Tubulin polymerization in the presence of myotoxin-3. Tubulin polymerization assay was performed in the absence (vehicle, red line) or in the presence of 15 µM myotoxin-3, either native (peak 1) or synthetic (oxidized and non-oxidized forms). Paclitaxel (15 µM) was used as a positive control. The increase in the turbidity at 350 nm is proportional to tubulin polymerization. The data shown are from a representative experiment of two performed.

**Figure 6 molecules-27-08241-f006:**
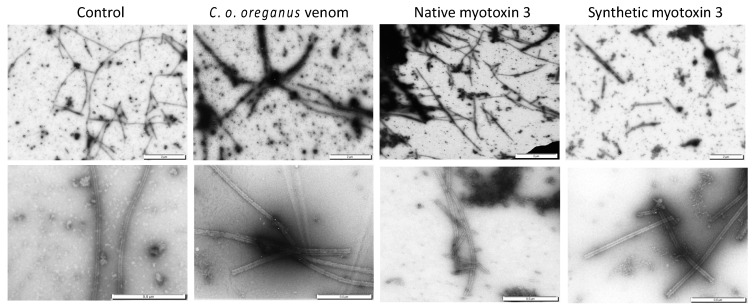
Microtubules assembled in the presence of myotoxin 3 and *C. o. oreganus* venom. Tubulin was allowed to polymerize at 37 °C in the absence (control) or in the presence of crude venom, native or synthetic myotoxin-3. Microtubules were then stained and observed by electron microscopy. Scale bars = 2 µm (upper panels), 0.5 µm (lower panels).

**Figure 7 molecules-27-08241-f007:**
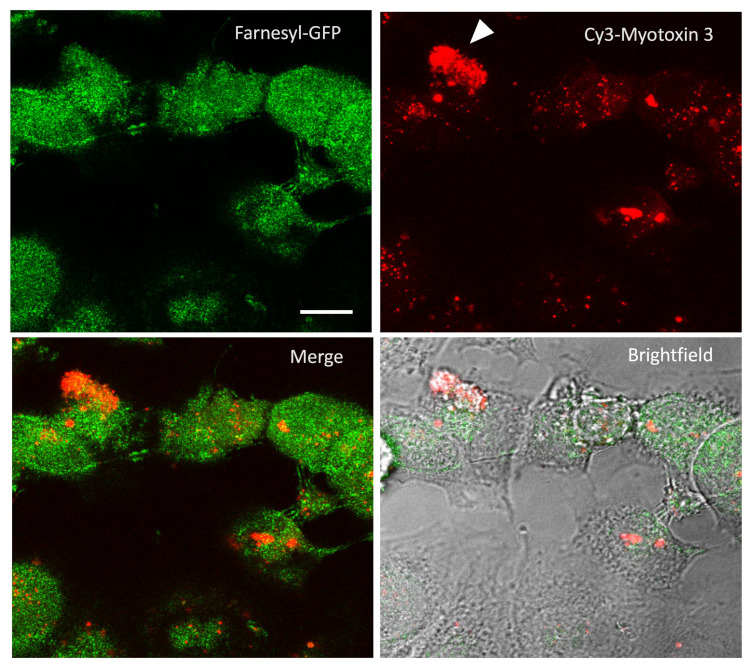
Cy3-myotoxin-3 uptake by MCF7 cells. MCF-7 cells were transiently transfected for farnesylated-GFP expression to delimit the cell membrane. Cells were then incubated for 1 h at 37 °C with labeled Cy3-myotoxin-3 (45-residue isoform, native). After low pH removal of cell-surface-bound peptide, cells were fixated. Images captured by confocal microscopy are presented as a stack of 30 confocal slices. Scale bar = 10 µm.

**Figure 8 molecules-27-08241-f008:**
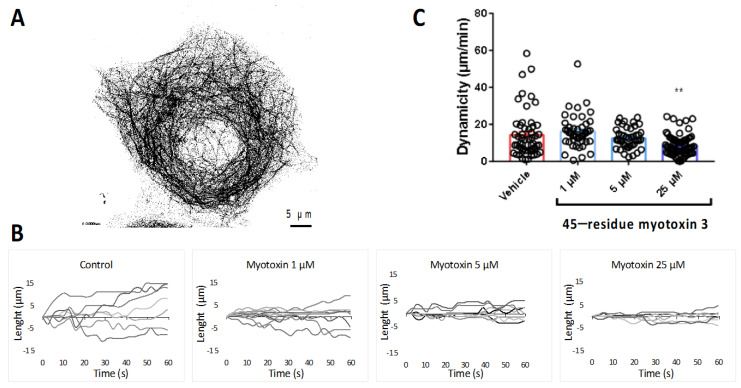
Microtubule dynamics in MCF7 cells treated with myotoxin-3. (**A**) MCF7 cell expressing EGFP-α-tubulin showing labeled microtubules. (**B**) Life history plots of the length changes of 7 microtubules in living MCF7 cells treated or not (control) for 3 h with 45-residue myotoxin 3, as described in the experimental procedure section. In the presence of myotoxin-3, MTs are characterized by less extensive growth and shortening events compared with control cells. (**C**) Overall dynamicity after treatment for 3 h without (control) or with 45-residue myotoxin 3. The data shown are from a representative experiment of two performed. ** *p* = 0.0013 relative to control.

**Figure 9 molecules-27-08241-f009:**
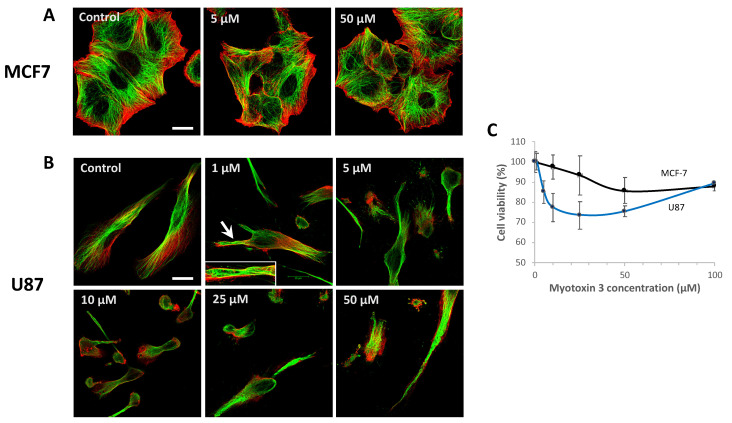
Microtubules and actin fibers labeling in MCF-7 and U87 cells. MCF-7 (**A**) and U87 (**B**) cells were treated for 24 h with the indicated concentration of myotoxin-3. Cells were then fixated, and actin fibers were labeled with phalloidin-TRITC (red) and microtubule network with anti-α-tubulin antibody (green). The arrow shows microtubules organized in wavy bundles (magnified in the inset). Bars = 20 µm. (**C**) The viability of MCF-7 and U87 cells was assessed by MTT after incubation with myotoxin 3 for 24 h at 37 °C. Data are from a representative experiment of two performed in triplicate.

**Figure 10 molecules-27-08241-f010:**
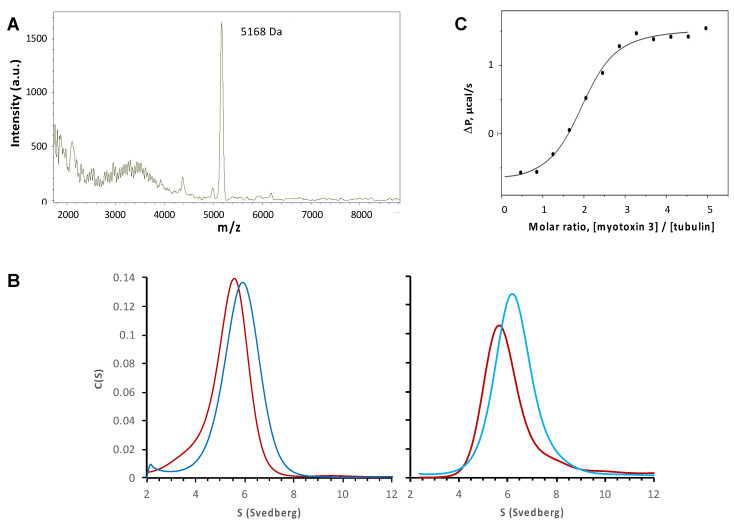
Binding of myotoxin-3 to microtubules and ab-tubulin dimers. (**A**) MALDI MS spectrum of peptides co-sedimented with microtubules. (**B**) Sedimentation profiles were obtained by analytical ultracentrifugation of tubulin alone (red) or tubulin with 45-residue myotoxin-3 (dark blue, left panel) or 43-residue myotoxins 3 (light blue, right panel). (**C**) The fitting curve of experimental data from titration of 55 µM tubulin by 2.2 mM of myotoxin-3 obtained by ITC. Data shown are from a representative experiment of two (**B**) or three (**A**,**C**) performed.

**Table 1 molecules-27-08241-t001:** Effects of myotoxin-3 on the parameters of microtubule dynamic instability in MCF7 cells.

			45-Residue Myotoxin 3		
Variables		CONTROL	1 µM	5 µM	25 µM
**Rate (µm/min)**	**G** **S**	25.9 ± 1.8 31.1 ± 2.6	26.7 ± 1.2 32.7 ± 2.0	26.9 ± 1.3 31.4 ± 1.9	20.3 ± 0.7 ^B^ 24.5 ± 1.3 ^B^
**Length changes (µm)**	**G** **S**	1.87 ± 0.16 1.59 ± 0.08	1.50 ± 0.06 ^C^ 1.71 ± 0.07	1.47 ± 0.06 ^C^ 1.52 ± 0.05	1.12 ± 0.04 ^A^ 1.19 ± 0.04 ^A^
**% time spent**	**G** **S** **P**	28.7 ± 2.1 17.3 ± 1.4 53.6 ± 2.7	29.2 ± 1.7 22.9 ± 1.6 ^B^ 47.9 ± 2.8	24.2 ± 1.5 20.0 ± 1.2 55.8 ± 2.2	20.5 ± 1.4 ^A^ 17.7 ± 1.0 61.9 ± 2.0 ^B^
**Transition frequency (events/min)**	**C** **R**	4.65 ± 0.42 18.91 ± 0.89	6.49 ± 0.52 ^B^ 20.08 ± 0.71	5.01 ± 0.27 20.43 ± 0.55	4.81 ± 0.31 20.24 ± 0.59
**Transition frequency (events/µm)**	**C** **R**	0.75 ± 0.08 0.93 ± 0.13	0.63 ± 0.04 0.74 ± 0.07	0.77 ± 0.10 0.95 ± 0.15	1.07 ± 0.11 ^C^ 1.58 ± 0.19 ^B^
**Dynamicity (µm/min)**		14.4 ± 1.73	16.01 ± 1.37	12.66 ± 0.86	8.65 ± 0.68 ^A^

Variables of microtubule dynamic instability were determined on living MCF7 cells after 3 h incubation with the indicated concentration of native myotoxin-3. G: growth; S: shortening; P: pause; C: catastrophe; R: rescue. Data are represented as mean ± SEM, n = 54 (control), n = 44 (1 µM and 5 µM) and n = 68 (25 µM), ^A^
*p* < 0.001, ^B^
*p* < 0.01, ^C^
*p* < 0.05, otherwise, statistically no significant.

## Data Availability

Not applicable.

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
