# Peer review of "Myotoxin-3 from the Pacific Rattlesnake Crotalus oreganus oreganus Venom Is a New Microtubule-Targeting Agent"

_molecules, 2022, doi:10.3390/molecules27238241_

Round 1

Reviewer 1 Report

The manuscript discusses the screening and further evaluation of a peptide with potential anti-microtubule activity. The presented work is interesting and enriches the current library of potential MTAs. Further, the work encompasses extensive isolation and dissecting of how the selected molecule might work. Overall the manuscript is well-written, and the structure is legible. Below, a few comments that may improve the manuscript are listed. 

General comments:

(1) Please simplify the writing. There are many complex sentences up to 2-3 lines long that may confuse the reader. The manuscript is already quite challenging to read due to the many detailed explanations and figures, thus simple sentences would be appreciated.

(2) In the introduction, the author swiftly described microtubules and their relevance in therapy. It is not clear how the microtubule networks behave in nature: what is the trigger for the switch between catastrophe and rescue? If this regulation involves MAPs, how does this protein group work? If the currently developed drugs could derive from both the antagonist and protagonist of microtubule network stability, which one is more effective/powerful/strategic or which one should be preferred at what stage (of a disease)? Does MTA have a specific target? How does it differentiate between normal and abnormal conditions? This general insight is useful for the reader to understand without having to read other publications.

(3) some figures are not proportional, please make sure that they are esthetically presented.

a few details:

line 48. "whose" refers to whom? mitosis or MTA or anticancer drug or the clinic?

line 49. colchicine was suddenly mentioned before being introduced. is it part of the example for the vinca-alkaloid?

line 51-54. an example for long sentences with subclauses that may confuse the reader. what is the subject of this sentence?

line 55. peptide-protein-enzyme are of one group: polypeptides. bioactive molecules encompass more groups, e.g. lipids, carbohydrates, secondary metabolites (alkaloids, statins, etc), and their derivatives. It is better to directly and specifically address 'peptides" or not classify them (since some of the drug examples at the end of the paragraph are small molecules - not peptides, proteins, or enzymes).

line 56. "It is thus ..". This sentence has no causative relation to the previous one.

line 79. oreganus is written twice (also in figure 1 legend), thus the sub-species is mentioned (differentiate it from e.g. C.o. helleri). Please be consistent about which one to use (C.o.o or C.o), for example in line 68.

Figure 1. the x-axis is in retention time but the explanation is in fraction numbers. please indicate the fraction (in the x-axis or mark the peak positions with arrows to indicate the fraction number). what is the vehicle? the sample was obtained in the RP column using ACN as the solvent, how to ensure that the ACN was removed after the sample was lyophilized? also, the fractions were eluted at different ACN concentrations, how would this affect the ACN content in the sample and further on the bioassay?

line 100 and figure 2. the scales are too small to read, please indicate which one is the "main" (presumably the one with the highest intensity?) and the inserted figures are the zoom-in of which peaks. 

line 115. the identified peptide is presumably highly cationic. does this reconcile with the elution profile in the preparative HPLC?

line 181. how was the reducing condition maintained in the reaction? was the reducing agent concentration higher? how would this affect the activity? was it possible for a spontaneous S-S bond formation to take place in the assay?

line 343-347. The 3 "thus" indicate a sequential/consequential relationship for these sentences, which seems not true, especially when it actually refers to a compound with different activity and from a different species.

line 353. .. and described ?? 1990 ... (missing something)

line 399-400. MCF7 and U87 are two distinct types of cancer cell lines. what was the motivation to choose them? was it to decode different cancerous cell morphology? There are reports that relate metastasis in the brain and the breast cancer cells (involving beta-III tubulin or tubulin tyrosine ligase like-4), is it related to that?

Reviewer 2 Report

The authors have purified myotoxin-3 from Crotalus oreganus venom and have shown that it affects cells by inducing microtubule assembly and inhibiting microtubule instability.  

The experimental work is sound and the conclusions are logical, but the paper is very poorly written to the point where it appears to contradict itself.  The authors are strongly advised to get someone who speaks English well to re-write the paper for them.  Once it is re-written to be made comprehensible, then it would constitute a valuable contribution to our understanding of how snake toxins work, at least this particular one, which seems to operate by a unique mechanism.

Specific criticisms follow.  Most of them are indications where the authors have failed to make clear their thoughts and have not communicated well with the reader.  There are many errors in grammar that I have not pointed out.

Line 19.   Very few readers are likely to know that Crotalus oreganus is the Pacific rattlesnake.  The authors should make clear what organism they are using, so that the reader does not need to use Wikipedia to check.

Line 24.  “microtubules tubulin” is not a viable phrase and should be replaced by “microtubules and tubulin”.

Line 34.  “and yet dynamic complex” should be replaced by “and dynamically complex”.

Line 57.  “incredible” is not a scientific word.  It literally means “not to be believed”.  In this context, it should be replaced by “many”.  It is used elsewhere as well and should be replaced.

Line 66. “species families” should be replaced by “species and families”.

Line 70.  What does “in cellulo” mean? Perhaps a better phrase should be found.

Line 86.  “mains peaks” should be replaced by “main peaks”.

Lines 87-88.  A “peak” does not reduce lag time etc.  What is inside the peak does.  This should be made clear.

Line 91. “venom” does not need to be italicized.  

Line 119.  Now that the disulfide bonds are mentioned, they should be identified.  Precisely what are the positions of the cysteine resides that they link?  This is not indicated in Figure 2C.  Perhaps that would be a good place to do that.  If the positions of the disulfides is not known, that should be stated.

Line 130. “seventh unidentified residues” should be “seventh unidentified residue”.

Line 136.  Suddenly “helleramine” is brought up.  What is it and where does it come from?  Why was it used? This needs to be made clear.

Line 140.  What does the phrase “Leucine 5 residues” mean?  Does it mean leucines at position 5 or 5 residues of leucine?  This should be made clear.

Lines 152-153.  How does the presence of a histidine residue lead to a decrease in molecular weight of 24 daltons?  This should be made clear because it seems contradictory.

Line 185 and elsewhere.  The authors refer to “vehicle”.  They need to define exactly what the vehicle is.

Figure 7.  The Y-axis in the graphs is given as “distance”. Does this mean “length”, as in microtubule length?  If so, that should be made clear.

Line 248 and elsewhere.  The authors talk about “representative” microtubules.  What exactly does this mean and how were they selected as “representative’?

Lines 292-293.  The phrase “were no more noticeable” is awkward and implies that both microtubules and actin have disappeared.  Is that what the authors intend to say?  More importantly, this seems to contradict the observation that myotoxin-3 induces microtubule assembly.  One would expect there to be more microtubules, not to have them disappear.  Also, why would actin be involved?  The authors should make the connection.

Line 338.  The authors give a Kd for myotoxin-3 and tubulin.  They say that it is comparable to those of other MTAs.  They should give the Kds of other MTAs to convince the reader that they are indeed comparable.

Line 351.  The authors state that the toxin is able to bind to tubulin and “to interfere with microtubule formation.”  “Interfere” means to block, whereas the authors present evidence that the toxin promotes microtubule assembly.  This needs to be re-worded.

Line 390.  Once again we have the apparent contradiction: the toxin that promotes microtubule assembly “causes “ disappearance of the microtubular network”.  This needs to be re-phrased.

Lines 412-415.  The authors speculate about the toxin interacting differently with different tubulin isotypes.  This is very reasonable and indeed probable, but their arguments here are not well-framed.  They state that myotoxin-3 may target an isotype predominantly expressed in neuron (should be “neurons”), and that this is why glioblastoma cells are more sensitive to the toxin than are MCF-7 cells.  The problem here is that glioblastoma cells arise from glial cells, not from neurons, and the tubulin isotype of glial cells is different from that of neurons. For example, the beta-3 isotype is predominantly (almost entirely) found in neurons and is not present in glial cells.  Thus the argument of the authors’ makes no sense.  They should probably drop that part of the argument, but, if the data is available in the literature, their interesting speculation about isotype specificity of myotoxin-3 could be illuminated by presenting the isotype compositions of glioblastoma and MCF-7 cells.

Lines 430-437.  The authors present a very interesting speculation that the cationic nature of myotoxin-3 may allow for binding to tubulin’s highly anionic C-termini.  They propose that this is analogous to the binding of tau and MAP7 to these regions.  The authors may want to compare the sequences of these MAPS to that of myotoxin-3 to see if this makes sense.  One might wonder why they mention MAP7 instead of the much better-known MAP2.

One final point.  Some of the toxins that bind to tubulin, namely, phomopsin, rhizoxin and halichondrin, are modified peptides.  Although their action is to inhibit microtubule assembly, the authors may want to see if there is any resemblance in their sequence of amino acids to any part of myotoxin-3.
